# Parents' knowledge and practices of child eye health care: A scoping review

**Nor Diyana Hani Ghani**[ID], **Norliza Mohamad Fadzil**[ID]*, **Zainora Mohammed**[ID], **Mohd Harimi Abd Rahman**[ID], **Normah Che Din**[ID]

Faculty of Health Science, Optometry and Vision Science Program, Centre for Rehabilitation and Special Needs Study (iCaRehab), University Kebangsaan Malaysia, Bangi, Malaysia

☯ These authors contributed equally to this work.
* norlizafadzil@ukm.edu.my

## Abstract

### Background

Treating children's eyes is not just a privilege for a child, it is an essential requirement for their visual health. Parents, as caregivers, have a paramount responsibility to make decisions regarding their children's eye health. Thus, this review aims to identify and summarise published information about parents' knowledge and practices on children's eye health care.

### Methods

Relevant articles searches were performed through a systematic search of databases (EBSCOhost, PubMed, and Scopus) using the keywords 'knowledge', 'practice', 'parent', 'eye', 'problem', and 'children'. This review was conducted and reported in line with the PRISMA-ScR. The methodological quality of the listed studies was assessed using A Modified McMaster Critical Review form based on the total score.

### Result

From a total of 235 studies retrieved through literature review and pearling, 219 remained after removing duplicates. After screening titles and abstracts, 204 irrelevant studies were excluded, leaving 15. After a detailed full-text review, four studies were excluded due to not meeting inclusion criteria. Thus, this review includes the remaining 11 studies. All eleven studies (n = 11) show that parents's knowledge and practices vary. Some parents display good knowledge regarding children's eye health care, for example, understanding the importance of wearing spectacles, the significance of children having normal vision, and where to seek eye examinations. Five studies (n = 5) showed that parents have good practices such as consulting doctors and seeking eye examinations and treatment at the hospital. Six studies (n = 6) showed that parents have misconceptions regarding knowledge, practices, and treatment of children's eye health.

**Data Availability Statement:** Minimal data set are within the manuscript.

**Funding:** The funder for this research is from Universiti Kebangsaan Malaysia research grant

(GUP-2020-086). The funders had no role in study design, data collection and analysis, decision to publish, or preparation of the manuscript.

**Competing interests:** No authors have competing interest

## Conclusion

This scoping review found that parents' knowledge and practices regarding children's eye health are poor. Parents' perceptions and practices about the cause and treatment of eye problems were tainted with misconceptions. Therefore, implementing structured programs to enhance awareness and promote the adoption of healthy practices for children's eye health is required.

## Introduction

Vision impairment is one of the most critical issues affecting anyone of any age, social status, or ethnic background [1]. Among the 36 millions individuals worldwide affected by vision impairment as reported by the World Health Organization (WHO), 29% are children and this figure will rise [2]. In Malaysia alone, 12.5% of preschoolers were reported to have vision impairment, of which 59.1% had moderate vision impairment and 61% had bilateral vision impairment [3].

According to the WHO Global Action Plan (2014–2019), early detection of eye problems is key to reducing 80% of the causes of vision impairment in children [4]. Vision problems such as amblyopia, strabismus, and uncorrected refractive errors can cause suppression, leading to difficulties in tasks such as writing, reading, and retaining information. Thus, delaying treatment not only harms children's development but also undermines their learning abilities, communication skills, overall health, career prospects, and quality of life [5,6]. Parents as the primary guardians of their children are responsible for ensuring timely access to eye care services [7,8]. The aspect of parental knowledge and practices concerning child eye health care has recently received significant attention [9,10]. However, some parents are not able to notice their child's vision impairment since some eye conditions are asymptomatic and children are frequently not aware of their poor vision [11,12].

Parents' knowledge and practices regarding children's eye care vary across different countries, influenced by factors such as education level, occupation, economic status, and cultural backgrounds [7,9,11,13–22]. It is evident from several studies that parents generally have some knowledge of children's eye care, though the level of this knowledge varies [11,16–19]. However, despite this awareness, a significant portion of parents fail to schedule eye examinations or follow-ups for their children [14,23]. Common factors cited include time constraints, forgetting about referrals, believing their child doesn't have eye problems, or considering them too young for thorough examinations [14]. Additionally, other studies highlight factors such as stigma, belief, ethnicity, and religious bias contributing to misconceptions of parents' knowledge and practices [7,9,13,17,20,21].

Therefore, it is crucial to understand parents' knowledge and practices regarding children's eye health. This understanding can provide an initial step in implementing effective intervention programs to enhance parents' knowledge and practices related to children's eye health care. An intervention conducted on parents to identify and address barriers to eye care resulted in significant improvement in adherence to eye care from less than 5% to a notable 59.2%. However, this effort comes with additional costs [23]. This study aimed to identify, summarise, and report the current knowledge and practice of parents regarding child eye health care. The findings of this study can serve as a foundation for implementing impactful awareness or intervention programs that are cost-effective for parents and guardians.

## Materials and methods

### Study design

This scoping review process was conducted according to the published methodological guidelines of evidence synthesis and framework [24–29]. The Preferred Reporting Items for Systematic Review and Meta-Analysis Extension to Scoping Review (Prisma-ScR) guidelines were used in this review [30]. This scoping review process was implemented by following these steps: 1) identifying the research question, 2) identifying relevant studies, 3) selecting studies, 4) charting the data and collecting, and 5) collating, summarising, and reporting the findings [24,26–29].

**Stage 1: Identifying the research question.** This scoping review was a component of a larger research study designed to develop and evaluate a mobile app regarding children's eye health for parents/guardians. The purpose of this scoping review was to identify and summarise what is reported in the literature regarding parents' knowledge and practices on children's eye health care. Specifically, the review aims to answer the following questions: "What is the current knowledge of parents regarding children's eye health care?" and "What are the current practices of parents regarding children's eye health care?"

**Stage 2: Identifying relevant studies.** To address the research question, a comprehensive literature search was conducted using three primary databases: EBSCOhost, PubMed, and Scopus, from August 2021 to July 2022, with an update in June 2023. The search utilized terms, truncation symbols, and Boolean operators based on the crucial components of the research question: problem, concept, and context (PCC). All keywords were determined through discussion among authors, including the experts from optometry (Table 1). Two search strings were finalized after several revisions and modifications (Table 2).

**Stage 3: Selecting a study.** The selection of studies for this scoping review followed the criteria outlined in Table 3. The search outcome was narrowed down through screening titles and abstracts to determine the eligibility of each study, followed by full-text screening [31]. Studies assessing parents' knowledge and practices of children's eye health care and services were included, while studies assessing the outcome of children's eye health care were excluded from this study.

The search was not limited by the study design, intervention, comparison, or other factors. In addition to the initial search, further relevant papers were identified through a method known as "pearling" which involves reviewing the reference lists of identified studies to find any additional relevant studies that might not have been sourced through the literature search. A modified McMaster Critical Appraisal form to assess 16 methodological quality items relating to the study purpose, literature review, study design, sample size, outcome, intervention, result, and conclusion was used [32].

In this review, the intervention component was not included as it was not the study's purpose. The form uses a rating scale where 'yes' earns 1 point, while 'no' or 'not applicable' (N/A) earns 0 points. This binary scoring system simplifies the evaluation process by providing clear

**Table 1. List of keywords and synonyms generated as search terms.**

| Knowledge | Practice | Parent | Eye | Problem | Children |
|---|---|---|---|---|---|
| Awareness | Idea | Guardian | Vision | Health | Child |
| Perception | Belief | Caregiver | Visual | Care | Pediatric |
| Attitude | | Caretaker | Ocular | Disease | School kid |
| Education | | | | Disorder | Youth |
| | | | | Examination | Preschool |
| | | | | Treatment | |

**Table 2. List of search strings.**

| | |
|---|---|
| **Search String 1** | **(Knowledge OR perception OR awareness OR attitude OR education\*) AND (parent\* OR guardian\* OR caregiver\* OR caretaker\*) AND (Eye\* OR vision\* OR visual\* OR ocular\*) AND (Problem\* OR health\* OR care\* OR disease\* OR examination\* OR treatment\*) AND (Children\* OR child OR pediatric OR school kid\* OR youth\* OR preschool\*)** |
| **Search String 2** | (practice\* OR idea\* OR belief\*) AND (parent\* OR guardian\* OR caregiver\* OR caretaker\*) AND (Eye\* OR vision\* OR visual\* OR ocular\*) AND (Problem\* OR health\* OR care\* OR disease\* OR examination\* OR treatment\*) AND (Children\* OR child OR pediatric OR school kid\* OR youth\* OR preschool\*) |

criteria for assessing the suitability of the studies. An overall score for the selected studies was calculated by adding up the points earned across all assessed criteria. Converting the total score into a percentage provides a clear and intuitive way to understand the overall quality of the study, where a higher percentage indicates a higher quality and a lower percentage suggests potential methodological weakness or limitation.

To ensure the reliability and consistency of the evaluation process, two reviewers (ZM and HAR) independently reviewed the full text of each abstract that was selected for inclusion. Two reviewers were selected to reduce the possibility of biases or errors arising from a single reviewer. Any discrepancies or disagreements between the reviewers regarding the evaluation of studies were resolved through the discussion.

**Stage 4: Charting the data.** A data extraction table has been designed to compile and report pertinent data by using a combination of table and text, aligned with the objective of this scoping review. Each full-text article was reviewed, and information was gathered from the data extraction process, including the author(s), the year of publication, and the mode of data collection. Data regarding the parents' knowledge and practices regarding children's eye health care were extracted from the selected studies if mentioned by the authors and relevant to the objective of the scoping review. Excel spreadsheets and Microsoft Word were used to organize and monitor the relevant data.

**Stage 5: Collecting, summarising, and reporting the results.** This scoping review involved the development of customized data charts to retrieve all information regarding studies, methods, and the overall result. An external reviewer (NCD) initially examined the forms to ensure that all necessary data was gathered and any feedback was integrated into the final version. Once agreement on included research and data extraction was achieved, Microsoft Excel was used as the data management software and compiled into a single database. The checklist was completed using the Preferred Reporting Items for Systematic Reviews and Meta-Analyses extension for Scoping Review (PRISMA-ScR) guidelines [30].

# Results

## Literature search

The search outcomes yielded 235 studies, 211 through the literature search and 14 from Pearling. After eliminating duplicates, the number of studies was reduced to 219. Following the PRISMA

**Table 3. Criteria selection.**

| Criteria | Inclusion | Exclusion |
|---|---|---|
| Publication Timeline | 2010–2023 | - |
| Document Type | Article (with empirical data) | Conference proceeding, chapters in the book, book series, books, reviews, etc |
| Language | English | - |
| Nature of study | No limitation | - |

ScR guideline, only studies relevant to the research question were included while screening titles and abstracts. Upon examining titles and abstracts, the majority (204) of the studies were excluded, leaving 15 studies for thorough evaluation through full-text review. Closer scrutiny revealed that four articles did not meet the research questions and objectives, resulting in a final selection of 11 studies for inclusion in this scoping review. The decision to exclude the four studies was based on their irrelevant content, which included studies on barriers, factors associated with eye care, and the development of the questionnaire. Fig 1 shows the flow diagram based on the Preferred Reporting Items for Systematic Reviews and Meta-Analyses extension for Scoping Reviews (PRISMA-ScR). This ensures transparency and strictness in the selection and evaluation of studies, enhancing the reliability and validity of the findings of this scoping review.

## Methodological quality

The Modified McMaster Critical Review form was used to summarise the evidence levels and critical appraisal scores of all included studies (Table 4) [29]. The methodological quality of the included studies ranged from moderate to excellent, with a score ranging from 6 to 11 out of a possible 11. The studies had a mean score of 9.4 ± 1.5 SD. All studies were cross-sectional, had a clear purpose, and had relevant literature. Most of the studies report descriptive findings.

## Study characteristic

All the studies selected aimed to identify the parents' knowledge and the current practices of parents regarding children's eye health care and were published between 2010 and 2023. Table 5 summarises the mode of data collection, knowledge, and practices among parents reported in the 11 selected studies. The type of studies were quantitative methods using questionnaires (n = 7); qualitative methods which involved focus group discussion or an in-depth interview (n = 3) and a combination of quantitative and qualitative methods (n = 1). The majority of the studies were conducted in developing countries–India [13,16], Nigeria [7,9,21],

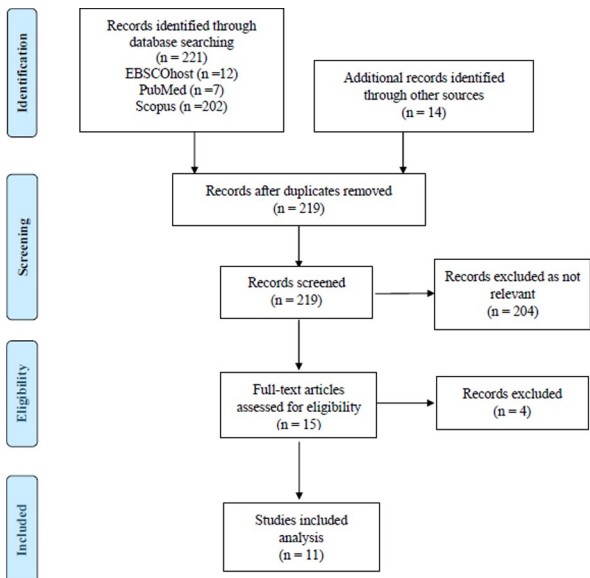

**Fig 1. Flow diagram of Preferred Reporting Items for Systematic Reviews and Meta-Analyses extension for Scoping Reviews (PRISMA-ScR).**

**Table 4. Levels of evidence and critical appraisal scores using the Modifeied McMaster Critical Review form.**

| Study | 1 | 2 | 3 | 4a | 4b | 4c | 5a | 5b | 7a | 7b | 7c | 7d | 8 | Total score /11 | Total score in % |
|---|---|---|---|---|---|---|---|---|---|---|---|---|---|---|---|
| Surrati et al., 2022 [15] | Y | N | CS | 553 | Y | Y | Y | Y | Y | Y | Y | NA | Y | 10 | 91 |
| Baashar et al., 2020 [11] | Y | N | CS | 97 | N | NA | Y | Y | Y | Y | NA | NA | Y | 6 | 55 |
| Ravindran et al., 2020 [16] | Y | Y | CS | 324 | N | Y | Y | Y | Y | Y | Y | Y | Y | 10 | 91 |
| Bruce et al., 2018 [14] | Y | Y | CS | 20 | Y | Y | Y | Y | N | Y | N | NA | Y | 8 | 73 |
| Donaldson et al., 2018 [20] | Y | Y | CS | 1317 | Y | Y | Y | Y | Y | Y | Y | Y | Y | 11 | 100 |
| Sukati et al., 2018 [19] | Y | Y | CS | 173 | Y | Y | Y | Y | Y | Y | Y | NA | Y | 10 | 91 |
| Ebeigbe & Emedike, 2017 [9] | Y | Y | CS | 35 | Y | Y | Y | Y | N | Y | Y | NA | Y | 9 | 82 |
| Alrasheed et al., 2016 [18] | Y | Y | CS | 387 | Y | Y | Y | Y | N | Y | Y | Y | Y | 10 | 91 |
| Amiebenomo et al., 2016 [7] | Y | Y | CS | 468 | Y | Y | Y | Y | Y | Y | Y | Y | Y | 11 | 100 |
| Senthilkumar et al., 2013 [13] | Y | Y | CS | 35 | Y | NA | Y | Y | N | Y | Y | NA | Y | 8 | 73 |
| Ayanniyi et al., 2010 [21] | Y | Y | CS | 1393 | Y | Y | Y | Y | Y | Y | Y | NA | Y | 10 | 91 |

McMaster items to be scored: 1- Was the purpose stated clearly? 2- Was relevant background literature reviewed? 3- What is the study design? 4a- Sample number; 4b- Was the sample described in detail?; 4c- Was the sample size justified?; 5a- Were the outcomes measurable and reliable?; 5b- Was the outcome valid?; 7a- Results were reported in terms of statistical significance.; 7b- Were the analysis method/s appropriate?; 7c- Clinical importance was reported? 7d- Drop-outs were reported? 8- The conclusion was appropriate given the study methods and result?. Y = yes, N = no, NA = not address, CS = cross-sectiol.

Saudi Arabia [11,15], Sudan [18], Swaziland [19], and only two (n = 2) studies were undertaken in a developed country (England) [17,20].

## Parents' knowledge

All studies included in this review revealed that parents are knowledgeable regarding children's eye problems. Among them, four studies (n = 4) found that more than 50% of parents have knowledge about children's eye problems [16,18,20,21], while another four studies (n = 4) reported less than 50% of parents have knowledge [7,11,15,19]. One study (n = 1) reported the percentage of parents who have knowledge about children's eye problems ranged from 20% to 85%, depending on specific eye problems [9]. Two studies (n = 2) noted that parents have knowledge without providing specific percentages [13,17].

Additionally, three other aspects of parents' knowledge regarding children's eye health discussed in the selected studies are knowledge of the importance of normal vision (n = 3) [9,19,21], knowledge of the importance of eye examinations and where to seek them (n = 7) [7,11,15,16,19–21] and knowledge of the benefits of eye treatments (n = 4) [9,13,17,19].

## Parents practice

Out of eleven included studies, only eight (n = 8) indicated that parents practice positive attitudes [7,9,11,15–18,20]. Five studies (n = 5) reported that parents seek a proper eye examination [7,9,11,15,20], four studies (n = 4) reported that parents seek information about children's eye health care from various sources [11,15,17,18] and five studies (n = 5) reported that parents are willing to allow their children to wear spectacles and consider necessary surgery [7,11,15,16,19].

Overall, this review reveals that parents have misconceptions and stigma about children's eye problems [9,13,17–19], practices [7,9,11,13,19,20] and treatments [9,13,17,20].

## Discussion

The objective of this scoping review was to identify and summarise the literature concerning parents' knowledge and practices regarding children's eye health care. Previous studies on this topic have been limited on a global scale. To our knowledge, this is the first scoping review to

**Table 5. Characteristic of studies on parental knowledge and practices regarding children's eye health.**

| No | Study | Mode of Data Collection | Knowledge | Practice |
|---|---|---|---|---|
| 1 | Surrati et al., 2022 [15] | Questionnaire | • 3.6% of parents have excellent knowledge about eye diseases.<br>• 18.2% of parents have good knowledge about eye diseases.<br>• 78.2% of parents have poor knowledge about eye diseases.<br>• 48.5% of parents know that annual eye examinations are beneficial for children. | • 58.6% of parents seek eye treatment from an ophthalmology clinic.<br>• Parents seek information from doctors, internet browsers, social media, and friends.<br>• 76.9% of parents allowed their children to wear spectacles.<br>• 84.5% of parents allowed their child to undergo surgery. |
| 2 | Baashar et al., 2020 [11] | Questionnaire | • The level of knowledge of parents was below average (2.03 ± 1.692) with a maximum score of 7.<br>• 52.6% of parents did not know the need for eye examinations for their child.<br>• 61.9% of parents believe there was no need for eye examinations. | • 52.6% of parents took their children for eye examinations.<br>• 12.4% of parents consult a doctor for eye problems.<br>• Parents seek information from<br>  • family doctor (37.1%)<br>  • social media (22.7%)<br>  • friends (14.4%)<br>• 55.7% of parents did not allow their children to wear spectacles.<br>• 71.1% of parents allowed their children to undergo surgery. |
| 3 | Ravindran et al., 2020 [16] | Questionnaire | • 89.5% of parents know about their child's eye problems.<br>• 98.2% of parents know where to get an eye examination. | • 96.6% of parents did not consult eye hospitals or doctors.<br>• 89.5% of parents allowed their children to wear spectacles. |
| 4 | * Bruce et al., 2018 [17] | Semi-structured interviews (This is a qualitative study; percentages are not available). | • Parents know the benefit of wearing spectacles.<br>• Parents believe their child has good vision.<br>• Parents believe their child should not require spectacles. | • Parents seek advice from family or community members.<br>• Parents allow their children to wear spectacles if there is a family history and if they consider the vision test to be reliable.<br>• Parents do eye tests on their children at home by themselves.<br>• Parents avoided getting spectacles for females because of cosmetic reasons. |
| 5 | *Donaldson et al., 2018 [20] | Questionnaire | • 15% of parents know about the eye screening program.<br>• 78.6% of parents know where to go for an eye examination.<br>• 66% of parents know that children can have an eye examination.<br>• 15% of parents know that occasional eye turns in children between the ages of 1 to 7 are not normal.<br>• 48% of parents believe wearing glasses can make the eye stronger.<br>• 85% of parents believe that the school will do vision screening tests for all eye problems. | • Parents seek eye care if their children have:<br>  • poor vision (67%)<br>  • advice from healthcare providers or teachers (64%)<br>  • eye examination follow-up (53%)<br>  • double vision (52%)<br>  • headache (47%)<br>  • eye turn (43%)<br>  • family history of eye problems (40%)<br>  • poor concentration (22%)<br>  • poor school achievement (18%) |
| 6 | Sukati et al., 2018 [19] | Questionnaire | • 46.9% of parents know about childhood eye conditions.<br>• 28.9% of parents did not know about the need for eye examinations.<br>• 73.4% of parents know that poor vision can affect children's performance at school.<br>• 45.3% of parents know that eye diseases are the cause of poor vision.<br>• 73.2% of parents know the benefit of wearing spectacles.<br>• 18.3% of parents believe there is no need for eye examinations.<br>• 25.3% of parents believe watching television is the most common cause of poor vision. | • 60.1% of parents never take their children for an eye examination.<br>• 31.3% of parents seek information from medical doctors.<br>• 50.3% of parents allowed their children to wear spectacles. |

(*Continued*)

**Table 5.** (Continued)

| No | Study | Mode of Data Collection | Knowledge | Practice |
|---|---|---|---|---|
| 7 | Ebeigbe & Emedike, 2017 [9] | Focus group discussion and In-depth interview | • Parents know about children's eye problems: <br> • blurred vision (85.7%) <br> • cataract (74.3%) <br> • conjunctivitis (48.5%) <br> • itching or redness (74.3%) <br> • cross-eye (34.3%) <br> • long-sightedness (20%) <br> • short-sightedness (48.5%) <br> • strabismus (57.1%) <br> • 71.4% of parents know about the benefits of wearing spectacles. <br> • 51.4% of parents know about the association between poor school performance with eye problems. <br> • 34.3% of parents believe that vitamins such as vitamin A and vitamin C can treat eye problems. <br> • Parents believe the causes of refractive error are: <br> • Carbohydrates (31.4%) <br> • hygiene (20%) <br> • night reading (40%) <br> • too much television (34.3%) <br> • 62.9% of parents believe good food is beneficial to eye health | • Parents bring their children for an eye examination if their children have: <br> • difficulty seeing the blackboard (48.5%) <br> • itching (24.8%) <br> • redness (25%) <br> • discharging (26.1%) <br> • squeezing eye (25) <br> • skipping line (22%) <br> • rubbing eye (20%) <br> • sitting too close to watch television (40.4%) <br> • cross-eye (20%) <br> • 51.4% of parents prefer drugs for eye treatment <br> • Parents prefer homemade and or local treatments: <br> • sugar solution (17.1%) <br> • coconut water or coconut leaves (34.3%) <br> • palm wine (14.3%) <br> • butter (28.6%) |
| 8 | Alrasheed et al., 2016 [18] | Focus group discussion and questionnaire | • 89.3% of parents know their children had refractive error by observation. <br> • 53.6% of parents believe that poor nutrition is the cause of refractive error in children. <br> • 45.6% of parents believe strabismus is untreatable. <br> • 14% of parents believe that traditional treatment is more effective for strabismus. | • 42.9% of parents visit eye specialists for eye problems. <br> • Parents seek treatment if children have: <br> • inflammation (21.4%) <br> • night blindness (14.3%) <br> • cataract (10.7%) <br> • strabismus (90%) <br> • 21.4% of parents seek traditional treatments. <br> • Parents seek information from: <br> • television (28.6%) <br> • health education workshops (25%) <br> • eye care provider (21.4%) <br> • radio (14.3%) <br> • school textbooks (10.7%) |
| 9 | Amiebenomo et al., 2016 [7] | Semi-structured questionnaire | • 19% of parents know about eye problems. <br> • 0.64% of parents know that a child should have a first eye examination at 6 months. <br> • Over 57% of parents believe that routine eye examination is not necessary. | • Over 60% of parents seek eye examinations for their children. <br> • Over 65% of parents allow their children to receive eye treatment. |
| 10 | Senthilkumar et al., 2013 [13] | In-depth interviews and focus group discussion (This is a qualitative study, percentages are not available). | • Parents know about common eye problems among children such as amblyopia and strabismus. <br> • Parents know the purpose of wearing spectacles. <br> • Parents believe refractive error will be reduced with spectacle corrections. <br> • Parents believe eye exercise or yoga is an alternative to spectacle corrections. <br> • Parents believe watching television, not playing outdoors, vitamin deficiency and poor food intake can cause refractive error. <br> • Parents believe strabismus is a sign of good luck and can cure itself with the child's growth. <br> • Parents believe healthy food can improve vision. <br> • Parents believe a nutritious diet, good sleep habits, restricted television, and oil baths can prevent eye problems. | • Parents avoided an eye examination because they feared that their child might need spectacles and stigma. <br> • Parents avoided getting spectacles for females because of cosmetic reasons. |

(*Continued*)

**Table 5.** (Continued)

| No | Study | Mode of Data Collection | Knowledge | Practice |
|---|---|---|---|---|
| 11 | Ayanniyi et al., 2010 [21] | Questionnaire | • 87.9% of parents know about the availability of eye care specialists.<br>• 89.35% of parents know that normal vision is very important. | • Parents seek treatment for eye problems from the following:<br>• Hospital (62%)<br>• Self-medicating (9%)<br>• Traditional medicine (8%) |

* developed country.

compile empirical studies focusing on parents' knowledge and practices regarding children's eye health care. To address this gap, a systematic literature search was conducted, and the related studies were thoroughly evaluated. Eleven studies were identified through a synthesis search using Preferred Reporting Items for Systematic Review and Meta-Analyses extension for Scoping Review (PRISMA-ScR) guidelines. These findings are crucial for identifying the current state of parents' knowledge and practices regarding children's eye health care, thereby facilitating the development of innovative and effective programs aimed at promoting proactive eye care-seeking behaviour among parents.

## Knowledge

The findings from this review highlight varying percentages of parents that have knowledge regarding children's eye problems across eleven selected studies. Four studies (n = 4) indicated that over half of the parents possess awareness regarding children's eye issues [16,18,20,21], whereas another four studies (n = 4) suggested that fewer than 50% of parents are informed about these matters [7,11,15,19]. The reason for the difference in the results of past studies can be due to the level of education, social and economic factors [16,18,20,21]. Studies suggest that parents with higher education tend to have a better understanding of children's eye problems because, with strong social connections and high-income, parents can easily access information, thereby automatically increasing their knowledge [9,13,15,18,19,21]. Parents also may acquire knowledge about children's eye problems through family who have experienced similar issues [16].

Among the aspects of parents' knowledge discussed in the selected studies is the understanding of the importance of normal vision [9,19,21]. They recognize that without normal vision, various aspects of their children's lives will be affected including learning abilities, communication skills, overall health, career prospects, and quality of life [5,33]. This understanding motivates parents to actively monitor their children's eye health and seek timely treatment if abnormalities are detected [4,20].

Based on the findings of some of the studies, most parents have the knowledge of the importance of eye examinations and where to seek them [7,11,15,16,19–21]. These parents are more likely to take proactive measures to ensure proper eye care for their children. This understanding helps parents prioritise children's eye examinations and reflects their commitment to safeguarding their children's eye health. By recognising the necessity of eye examinations, parents acknowledge their role in facilitating early detection and intervention [4,20].

Moreover, parents who are knowledgeable about the benefits of eye treatments for their children are empowered to make decisions regarding their care and are more likely to adhere to proper treatments [9,13,15,17,19]. This knowledge helps parents understand the necessity of seeking proper treatment options and weighing potential risks and benefits for their children's eye health care. Consequently, this knowledge increases the parents' adherence to

recommended treatments, ensuring their children receive proper eye care [33]. This suggests that when parents understand the benefits of treatments, it will enhance their ability to support their children's eye health and promote the delivery of timely care and effective interventions [9,13,15,17,19].

## Practices

In this review, more than half of the selected studies indicated that parents practice positive attitudes towards children's eye health care [7,9,11,15–18,20]. Five of the selected studies reported that parents seek proper eye examinations, such as consulting doctors, seeking eye examinations and treatment at the hospital [7,9,11,15,20]. Proper eye examinations are important for detecting any potential eye problems or abnormalities in children, which can prevent or minimize the progression of eye conditions and preserve vision [7,9,11,13,16,19,20]. Regular eye examinations also enable the monitoring of eye health and the identification of any changes over time. Furthermore, consulting healthcare professionals ensures that children receive accurate diagnoses and appropriate recommendations for their specific needs.

This review found that parents are proactive in seeking information about children's eye health care, utilizing a range of sources including doctors, eye practitioners, internet browsers, social media, family, friends, and others [11,15,18,19]. Consulting eye care professional provides them with accurate information and helps to get timely eye examinations [15,18]. Additionally, using online resources such as Internet browsers and social media platforms allows parents access to a wide range of information about children's eye health. This helps parents gain a better understanding of different aspects of eye health care and potential issues affecting their children's eyes.

Based on the findings, some parents expressed willingness to let their children wear spectacles [7,15,16,19], and were open to considering necessary treatments such as surgery [11,15]. This demonstrates a proactive approach to addressing children's vision needs. Their openness to treatment options reflects a prioritization of their children's eye health, ensuring optimal eye care for their children.

## Misconceptions

While many parents show that they have knowledge about children's eye health, this review also found that some parents demonstrate misconceptions about eye problems [9,13,17–19]. Some parents wrongly believe that only older people may have eye problems [19], or that poor nutrition is the primary cause of refractive error in children and some parents assume that the refractive errors do not lead to visual impairment [13,18]. Additionally, some parents assume that their children are too young for an eye examination [15,18,20]. These misconceptions lead them to overlook the necessity of routine eye examinations under the assumption that their children's vision is adequate or satisfactory [7,11,18]. Without appropriate knowledge of children's eye health, parents may not fully understand the implications of their decision, potentially leading to untreated eye problems.

Parents across both developed and developing countries exhibit similar levels of knowledge regarding children's eye health, yet still hold misconceptions and stigma, reveal an intriguing relationship between knowledge and stigma. This implies that knowledge alone does not necessarily translate into reduced stigma. Despite having knowledge, stigma can still persist due to cultural, societal, or psychological factors [11]. Parents who are more informed about the importance of normal vision, eye examinations, and treatments tend to have a better understanding and are less likely to hold stigmatizing beliefs. However, parents with lack comprehensive knowledge, can perpetuate misconceptions and stigma. When parents lack knowledge

about eye health, they might rely on misconceptions or outdated information, which can perpetuate stigma.

Some parents are also reported to have misconceptions regarding both the practices [7,9,11,13,19,20] and treatments [9,13,17,20] of children's eye problems. They believe that routine eye examinations are unnecessary [7,11,15,19]. As a result, they refrain from consulting doctors or seeking treatment for their children's eye health concerns. Some parents displayed a preference for alternative approaches, such as drugs, self-remedies, or seeking traditional treatment and medication for their children [9,18,21], because they believe that traditional medicine and exercise are more effective in treating eye problems such as strabismus [9,18]. Some parents refuse to allow their children to wear spectacles due to concerns about the cosmetic appearance and stigma. Parents think wearing glasses might make their children feel embarrassed or different from others [9,13,18]. Some parents believe that strabismus is a sign of good luck that will resolve itself when the child grows up [13] and they believe that poor nutrition and carbohydrates lead to refractive errors and that eating healthy foods such as carrots, fish, vegetables, and eggs can improve vision and cure eye problems [9,13,18]. These misconceptions may lead to delays in diagnosis and treatment and potentially increase incidents of eye problems in children.

In conclusion, this scoping review found that, although parents have the knowledge and positive practices regarding children's eye health care, there are still misconceptions about the knowledge, practices, and treatment. Therefore, reinforcing accurate knowledge becomes crucial to promoting precise knowledge and practices and improving children's eye health care. The implication of these misconceptions among parents can potentially affect parents' judgement on children's eye health care which may jeopardise children's development and overall health [5]. Thus, there is an evident need to enhance parents' knowledge and practice regarding children's eye health care. This can be achieved through the implementation of more informative, innovative, and interactive campaigns or educational programs aimed at increasing parents' knowledge and practices of children's eye care behaviour.

## Study limitations

Even though this review was conducted utilizing PRISMA-ScR guidelines and a methodological approach, some limitations of this study must be acknowledged. Only English-language scientific articles with full-text, abstract availability, as well as open-access articles were screened and included in this study. There is a possibility that some studies are missed. This study only utilized three search engines, namely EBSCOhost, PubMed, and Scopus potentially resulting in oversight of some relevant studies due to lack of accessibility. However, two searches were carried out on these three search engines until a point of saturation to reduce the possibility of this risk.

## Conclusion

### Implication for clinical practice

The study provides insights into parents' knowledge and practices regarding children's eye health care. The present findings suggest a need for better eye health education and communication on the importance of accessing eye care for young children among all parents in the communities. Therefore, pragmatic strategies to improve and redesign children's eye health awareness campaigns are necessary. Awareness campaigns about common eye problems, routine eye examinations, and the consequences of delayed eye examinations for children can be implemented through various channels such as billboards, mainstream media, and social media to promote better eye health practices among parents and caregivers.

### Implication for future study

Future research should concentrate on creating unique, engaging tools to educate parents on providing better eye care for their children. Boosting children's eye health will benefit the children as well as the community.

## Supporting information

**S1 Checklist. Preferred Reporting Items for Systematic reviews and Meta-Analyses extension for Scoping Reviews (PRISMA-ScR) checklist.**
(PDF)

## Author Contributions

**Conceptualization:** Nor Diyana Hani Ghani.

**Data curation:** Nor Diyana Hani Ghani.

**Formal analysis:** Nor Diyana Hani Ghani, Zainora Mohammed, Mohd Harimi Abd Rahman.

**Methodology:** Nor Diyana Hani Ghani, Norliza Mohamad Fadzil, Normah Che Din.

**Supervision:** Norliza Mohamad Fadzil.

**Writing – original draft:** Nor Diyana Hani Ghani.

**Writing – review & editing:** Norliza Mohamad Fadzil, Zainora Mohammed, Mohd Harimi Abd Rahman, Normah Che Din.

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
