## [Decision Letter · Decision Letter 0]

21 Feb 2024

PONE-D-23-40902Parents’ knowledge and practice of child eye health care: A scoping reviewPLOS ONE

Dear Dr. Mohamad Fadzil,

Thank you for submitting your manuscript to PLOS ONE. After careful consideration, we feel that it has merit but does not fully meet PLOS ONE’s publication criteria as it currently stands. Therefore, we invite you to submit a revised version of the manuscript that addresses the points raised during the review process. Please submit your revised manuscript by Apr 06 2024 11:59PM. If you will need more time than this to complete your revisions, please reply to this message or contact the journal office at plosone@plos.org. Please include the following items when submitting your revised manuscript:A rebuttal letter that responds to each point raised by the academic editor and reviewer(s). You should upload this letter as a separate file labeled 'Response to Reviewers'.A marked-up copy of your manuscript that highlights changes made to the original version. You should upload this as a separate file labeled 'Revised Manuscript with Track Changes'.An unmarked version of your revised paper without tracked changes. You should upload this as a separate file labeled 'Manuscript'.

We look forward to receiving your revised manuscript.

Kind regards,

Liat Gantz, PhD

Academic Editor

PLOS ONE

Journal Requirements:

"Universiti Kebangsaan Malaysia research grant (GUP-2020-086)"

"This work was supported by Universiti Kebangsaan Malaysia research grant (GUP-2020-086)"

"Universiti Kebangsaan Malaysia research grant (GUP-2020-086)"

4. We note that your Data Availability Statement is currently as follows: "All relevant data are within the manuscript."

5. Please ensure that you refer to Figure 1 in your text as, if accepted, production will need this reference to link the reader to the figure.

7. Please update your submission to use the PLOS LaTeX template. The template and more information on our requirements for LaTeX submissions can be found at http://journals.plos.org/plosone/s/latex.

**Additional Editor Comments:**

Your revised manuscript should include more detailed methodology specifying the McMaster Critical Appraisal Tool, its scoring system, the reasons for exclusion of a large number of studies identified and why only eleven were included in the final review. The results should detail specific questions or questionnaires that were included in the studies, and the Tables (specifically Table 5) should be modified. In addition, when reporting means, it would be advisable to also report the range of the scores. The discussion should include the association between knowledge and practice patterns, and the implications of the parents' knowledge to visual dysfunctions and their developmental, occupational, and economic consequences. It would be advisable to have your final draft edited by an English speaker to make sure the timing and the grammar are correct.

Reviewers' comments:

Reviewer's Responses to Questions

**Comments to the Author**

1. Is the manuscript technically sound, and do the data support the conclusions?

Reviewer #1: Yes

Reviewer #2: Partly

2. Has the statistical analysis been performed appropriately and rigorously? 

Reviewer #1: N/A

Reviewer #2: N/A

3. Have the authors made all data underlying the findings in their manuscript fully available?

Reviewer #1: No

Reviewer #2: Yes

4. Is the manuscript presented in an intelligible fashion and written in standard English?

Reviewer #1: Yes

Reviewer #2: No

5. Review Comments to the Author

Reviewer #1: This study identified and summarized what is reported in 233 the literature regarding parents’ knowledge and practice on children’s eye health care.

Abstract:

1. There is no data regarding the populations that were included at the articles that were included in the review.

2. How did the information was obtained from the parents – by questionnaire? By a valid one? Is it the same questionnaire in all studies that were included?

3. Age range of children that were addressed in parent's knowledge assessment is missing.

4. What key words were used for the systematic literature review? Any excluded studies and why?

Introduction:

1. Line 58: it is better do detail which vision impairments could cause the mentioned complications.

2. Please explain the mechanism in which vision problems and amblyopia development could cause have consequences later on developmental, educational, occupational consequences.

3. Line 67: stigma is not the main cause for low knowledge on children's eye problems and parents who do not seek for eye examination on time to their child. Please find additional main obstacles for parents who do not adhere to vision exams in the following study: https://pubmed.ncbi.nlm.nih.gov/35970195/

4. Please give examples to intervention programs that succeed in lowering the amblyopia rates and it's consequences following good parental knowledge and adherence to recommendations. Such as: USPSTF. The following articles may be helpful: https://aapos.org/members/vision-screening-guidelines , https://pubmed.ncbi.nlm.nih.gov/28873168/ , https://pubmed.ncbi.nlm.nih.gov/31514739/ , https://pubmed.ncbi.nlm.nih.gov/25647611/

Methods:

5. Please specify if the McMaster Critical Appraisal Tool is a software or what kind of tool.

6. There is no mention to this tool and number of studies that were excluded by this tool on the diagram of figure 1. Please specify.

7. Please specify which questionnaires were used to assess partn's knowledge. An example for such a questionnaire and its validity process can be found in the following study, that also should be mentioned in the discussion: https://pubmed.ncbi.nlm.nih.gov/37620666/

8. Figure 1: please detail regarding the reasons for the massive exclusion of papers. Was it done by the McMaster tool or by the author?

Results:

9. Lines 119-121: it is mentioned that this tool provides a dichotomous score (yes/no), but on lines 185-188 and on table 4 it gives a score of 6-11. Please explain.

10. Tables 4-5: very good tables. Please detail the title of table 5.

Discussions:

11. Please detail the main findings at the first paragraph.

12. Are there any other examples from other fields on health literacy and its effect on health status.

13. Please add additional reasons for parents who do not adhere to vision exams: https://pubmed.ncbi.nlm.nih.gov/35970195/

14. Line 320: no need for the "LIMITATIONS" title. It is part of the discussion.

15. Please explain why so many studies were excluded from the analysis. Could be that so many studies are with low quality regarding assessment of parent's awareness?

Reviewer #2: Dear Editor,

Thank you for giving me the opportunity to review the manuscript Parents’ knowledge and practice of child eye health care: A scoping review. The authors conducted a scoping review using the PRISMA-ScR guidelines with the following questions: “what is the current knowledge of parents regarding common eye problems in children?” and “what is the practice of parents regarding common eye problems in children.” This is an important topic and has the potential for making a significant contribution to the literature.

The authors did an excellent job of presented their methodology and results of the first three of the scoping review steps: 1. Identifying the research questions 2. Identification of relevant studies 3. Selecting the studies and appraisal using the Modified McMaster critical appraisal tool.

The manuscript requires more explanations regarding the results of the last two steps: 4. Charting the data and 5. Collecting, summarizing and reporting the results.

This is apparent in table 5 and in lines 208 to 230. Table 5 contains percentages regarding awareness and practice. However, it is not clear what those percentages refer to. For example, in paper 7, does 71.4% refer to the percentage of parents who “know about the benefit of wearing spectacles”? The authors must explain how they got the percentages in the methods and in the table legend.

In the paragraph describing parents’’ knowledge – how is the category “knowledgeable” defined? Are the authors using the percentages in table 5? If so, what is the cut-off? 50%? Same question regarding the categories “aware”, “did not have knowledge” and “misconceptions.”

The same questions apply to parents’ practice (lines 219-230). How were the categories “practice positive attitude” and “negative practice” defined? Was this based on the percentages in table 5? If so, when there is more than one statement, on which is it based?

In addition, it will make table 5 easier to understand if the authors color coded the awareness and practice columns. The authors should put an asterisk by studies carried out in developed countries.

Did the authors find an association between “knowledge” and “practice”?

Minor corrections:

1. The English in the introduction could be improved.

2. Line 118 has the word (name?) pearling in parentheses. What is this referring to?

3. Results lines 151-2. Please state why three were excluded? Due to which criteria?

4. Line 188 – only one numeral after the decimal place.

5. Table 4 – what do the percentages refer to?

6. In the discussion, starting line 249, the authors write “majority of parents have knowledge about children’s eye health care and common eye problem in children.” This cannot be concluded from the study. The scoping review found that in majority of studies parents have knowledge. We do not know if this can be extrapolated to the majority of parents.

7. It would be interesting to compare the findings from studies in developed countries to those in less developed countries.

I look forward to reading a revised version of the manuscript and believe it would have the potential to make an impact on children’s visual health.

6. PLOS authors have the option to publish the peer review history of their article (what does this mean?). If published, this will include your full peer review and any attached files.

Reviewer #1: No

Reviewer #2: No

---

## [Author Response · Author response to Decision Letter 0]

12 Jun 2024

Manuscript Comment and Correction

1 Please ensure that your manuscript meets PLOS ONE's style requirements, including those for file naming. The PLOS ONE style templates can be found at 

Thank you for the comment. This manuscript has been amended to follow PLOS ONE guidelines.

2 Thank you for stating the following financial disclosure: 

“Universiti Kebangsaan Malaysia research grant (GUP-2020-086)”

Please state what role the funders took in the study. If the funders had no role, please state: “The funders had no role in study design, data collection and analysis, the decision to publish, or preparation of the manuscript.”

Thank you for the comment. This statement has been amended and included in the online submission form and cover letter.

3 Thank you for stating the following in the Acknowledgments Section of your manuscript: 

"This work was supported by Universiti Kebangsaan Malaysia research grant (GUP-2020-086)"

"Universiti Kebangsaan Malaysia research grant (GUP-2020-086)"

Thank you for the comment. This statement has been removed from the manuscript.

4 We note that your Data Availability Statement is currently as follows: "All relevant data are within the manuscript."

Please confirm at this time whether or not your submission contains all the raw data required to replicate the results of your study. Authors must share the “minimal data set” for their submission. PLOS defines the minimal data set to consist of the data required to replicate all study findings reported in the article, as well as related metadata and methods (https://journals.plos.org/plosone/s/data-availability#loc-minimal-data-set-definition).

Thank you for the comment. The statement in the online submission form has been changed to the minimal data set are within the manuscript.

5 Please ensure that you refer to Figure 1 in your text as, if accepted, production will need this reference to link the reader to the figure.

Thank you for the comment. Figure 1 has been added to the text.

6 Please include captions for your Supporting Information files at the end of your manuscript, and update any in-text citations to match accordingly. Please see our Supporting Information guidelines for more information: http://journals.plos.org/plosone/s/supporting-information.

 Not applicable.

7 Please update your submission to use the PLOS LaTeX template. The template and more information on our requirements for LaTeX submissions can be found at http://journals.plos.org/plosone/s/latex.

Thank you for the comment. This manuscript has been amended to follow the PLOS LaTeX template.

Additional comments

8 Your revised manuscript should include a more detailed methodology specifying the McMaster Critical Appraisal Tool, its scoring system, the reasons for the exclusion of a large number of studies identified and why only eleven were included in the final review. 

Thank you for highlighting this issue. Detailed methods for McMaster Critical Appraisal Tools and its scoring system have been added in the methodology: selecting study section (lines 132-147).

Based on the PRISMA Scr guideline, during the titles and abstracts screening, only articles that are relevant to the research question are included. A large number of articles were excluded from this review because the title and abstract were not relevant. Explanation for this has been added in the result section (lines 174-181).

9 The results should detail specific questions or questionnaires that were included in the studies, and the Tables (specifically Table 5) should be modified. 

Thank you for the comment. In this review, seven studies used questionnaires, one study used in-depth interviews and three studies used mixed methods (questionnaire, interview, and focus group discussion). All the studies used different questionnaires, each of which had been piloted and validated. Among the questions asked regarding parents’ knowledge in the studies were eye care, education of the visually impaired, refractive error, and eye disease. 

Details regarding the mode of data collection used in the studies to obtain information from parents in each study have been added in Table 5 column 2.

10 In addition, when reporting means, it would be advisable to also report the range of the scores. 

Thank you for the comment. The range of the McMaster score in this review was 6-11. This statement has been added in the methodological quality section (lines 217-219).

-11 The discussion should include the association between knowledge and practice patterns, and the implications of the parent's knowledge to visual dysfunctions and their developmental, occupational, and economic consequences. 

Thank you for the comment. The association between knowledge and practice is discussed in lines 324-326 and the implications of parents’ knowledge have been included in lines 374-376.

12 It would be advisable to have your final draft edited by an English speaker to make sure the timing and the grammar are correct. 

Thank you for the comment. This manuscript has been sent for proofreading as suggested.

Reviewer 1

Abstract

1 There is no data regarding the populations that were included in the articles that were included in the review. 

Thank you for the comment. This review included all studies that were performed on parents with children aged from birth to 18 years old. 

2 How did the information was obtained from the parents – by questionnaire? By a valid one? Is it the same questionnaire in all studies that were included? 

Thank you for the comment. 

In this review, seven studies used questionnaires, one study used in-depth interviews and three studies used mixed methods (questionnaire, interview, and focus group discussion). All the studies used different questionnaires, each of which had been piloted and validated. 

Details regarding the mode of data collection used in the studies to obtain information from parents in each study have been added in Table 5 column 2.

3 Age range of children that was addressed in the parent's knowledge assessment is missing. 

Thank you for the comment. In this review, all studies performed on parents with children (aged from birth to 18 years old) were included.

4 What keywords were used for the systematic literature review? Any excluded studies and why?

Thank you for the comment. The keywords (lines 31-32) used for the literature search and the reason for excluded studies (lines 37-40) have been added to the abstract.

Introduction 

5 Line 58: it is better to detail which vision impairments could cause the mentioned complications. 

Thank you for the comment. The statement was quoted from the WHO, which did not specify the type of visual impairment. The definition of visual impairment in that article is “Vision impairment occurs when an eye condition affects the visual system and its vision functions.”

6 Please explain the mechanism in which vision problems and amblyopia development could cause have consequences later on developmental, educational, and occupational consequences. 

Thank you for the comment. The statement regarding the mechanism of vision problems that leads to developmental, educational, and occupational consequences has been added in the introduction (lines 66-68).

7 Line 67: stigma is not the main cause for low knowledge on children's eye problems and parents who do not seek for eye examination on time to their child. Please find additional main obstacles for parents who do not adhere to vision exams in the following study: https://pubmed.ncbi.nlm.nih.gov/35970195/

Thank you for highlighting this issue. True, stigma and belief are not the only reasons for low knowledge of children’s eye problems among parents. This paragraph has been amended accordingly (lines 73-82).

8 Please give examples of intervention programs that succeed in lowering amblyopia rates and it's consequences following good parental knowledge and adherence to recommendations. Such as: USPSTF. The following articles may be helpful: https://aapos.org/members/vision-screening-guidelines, https://pubmed.ncbi.nlm.nih.gov/28873168/ , https://pubmed.ncbi.nlm.nih.gov/31514739/ , https://pubmed.ncbi.nlm.nih.gov/25647611/

Thank you for the comment. An example of the intervention program that improved adherence to eye care has been added in the introduction (lines 77-80)

Method

9 Please specify if the McMaster Critical Appraisal Tool is software or what kind of tool. 

Thank you for the comment. McMaster Critical Appraisal Tools is a form that has a set of questions designed to assess the validity and reliability of the research study. In this review, this form was completed for all selected studies. 

 Detailed methods for McMaster Critical Appraisal Tools and its scoring system have been added in the selecting study section (lines 136-143).

10 There is no mention of this tool and the number of studies that were excluded by this tool on the diagram of Figure 1. Please specify. 

Thank you for the comment. McMaster Critical Appraisal Tools have been described in the selecting study section (lines 132-143).

McMaster Critical Appraisal Tools was conducted on the 11 studies that have been selected. Based on the evaluation, all the 11 studies were accepted. This was not included in Figure 1, it is not part of the PRISM Scr. 

11 Please specify which questionnaires were used to assess parent's knowledge. An example of such a questionnaire and its validity process can be found in the following study, which also should be mentioned in the discussion: https://pubmed.ncbi.nlm.nih.gov/37620666/

Thank you for the comment. 

In this review, seven studies used questionnaires, one study used in-depth interviews and three studies used mixed methods (questionnaire, interview, and focus group discussion). All the studies used different questionnaires to assess parents’ knowledge, each of which had been piloted and validated. 

Details regarding the mode of data collection used in each study to obtain information from parents have been added in Table 5 column 2.

12 Figure 1: please detail regarding the reasons for the massive exclusion of papers. Was it done by the McMaster tool or by the author?

Thank you for the comment. 

Figure 1 is the flow diagram for PRISMA ScR. PRISMA ScR was used for the methodology and reporting of this scoping review. 

Based on the PRISMA Scr guideline, during the titles and abstracts screening, only articles that are relevant to the research question are included. A large number of articles were excluded from this review because the title and abstract were not relevant. Explanation for this has been added in the result section (lines 174-181).

The exclusion of studies was done by the author based on PRISMA ScR guidelines. The McMaster Critical Appraisal Tools was used to evaluate the selected studies (after PRISMA ScR).

Result

13 Lines 119-121: it is mentioned that this tool provides a dichotomous score (yes/no), but on lines 185-188 and table 4 it gives a score of 6-11. Please explain. 

Thank you for the comment. McMaster Critical Appraisal Tools used dichotomous scores which yes= 1, and no= 0. The range of the McMaster score in this review was 6-11. This statement has been added in the methodological quality section (lines 217-219).

14 Tables 4-5: very good tables. Please detail the title of table 5. 

Thank you for the comment. The title for Table 5 has been revised to accurately reflect its contents.

Discussion

15 Please detail the main findings in the first paragraph. 

Thank you for the comment. The main findings have been discussed in a separate paragraph Knowledge and Practice 

16 Are there any other examples from other fields on health literacy and its effect on health status. 

Thank you for the comment. The statement regarding health literacy and its effect on health status has been added (lines 289-295 )

17 Please add additional reasons for parents who do not adhere to vision exams: https://pubmed.ncbi.nlm.nih.gov/35970195/

Thank you for the comment. The additional reason for parents who do not adhere to vision exam has been added (lines 309-314)

18 Line 320: no need for the "LIMITATIONS" title. It is part of the discussion. 

Thank you for the comment. The word “limitation” has been changed to “Study Limitation” following the latest Plos One manuscript format. 

19 Please explain why so many studies were excluded from the analysis. Could be that so many studies are with low quality regarding assessment of parent's awareness? 

Thank you for the comment. 

Based on the PRISMA Scr guideline, during the titles and abstracts screening, only articles that are relevant to the research question are included. A large number of articles were excluded from this review because the title and abstract were not relevant. Explanation for this has been added in the result section (lines 174-181).

Reviewer 2

1 Table 5 contains percentages regarding awareness and practice. However, it is not clear what those percentages refer to. For example, in paper 7, does 71.4% refer to the percentage of parents who “know about the benefit of wearing spectacles”? The authors must explain how they got the percentages in the methods and in the table legend. 

Thank you for the comment. The percentages in Table 5 refer to the percentage of parents. This has been amended for better understanding. The percentages were taken directly from the selected studies. 

2 In the paragraph describing parents’’ knowledge – how is the category “knowledgeable” defined? Are the authors using the percentages in table 5? If so, what is the cut-off? 50%? Same question regarding the categories “aware”, “did not have knowledge” and “misconceptions.” 

Thank you for the comment. The statement regarding knowledgable has been amended. Yes, the result discussed was based on the percentages in Table 5. 

3 The same questions apply to parents’ practice (lines 219-230). How were the categories 

---

## [Decision Letter · Decision Letter 1]

30 Jul 2024

PONE-D-23-40902R1Parents’ knowledge and practices of child eye health care: A scoping reviewPLOS ONE

Dear Dr. Mohamad Fadzil,

I commend the authors for their revised version of the manuscript which addressed most reviewer comments and also reads well. Reviewer 2 still requires clarification for Tables 4 and 5 and their description in the results section. In addition to the clarification, the legend accompanying Table 5 should be modified to describe the numbers provided in parentheses and what they represent.

Additionally, a few minor corrections would still benefit the manuscript as described below.We invite you to submit a revised version of the manuscript that addresses the points raised during the review process.

We look forward to receiving your revised manuscript.

Kind regards,

Liat Gantz, PhD

Academic Editor

PLOS ONE

Journal Requirements:

Additional Editor Comments :

Some minor additional comments that will improve the manuscript:

Abstract: parents have a “huge” responsibility – perhaps “paramount” is a better term?

Line 57: Not sure that the word “Shockingly” is required and appropriate for a scientific report. Perhaps better rephrased Amongst the 36 million individuals worldiwde affected by vision impairment as reported by the World Health Organization (WHO), 29% are children, and this figure will rise [2].

Line 230: that parents are knowledgeable

Study 4 does not have percentages at all.

Study 9, knowledge – “parents believe that routine eye exam is not necessary – this has no score.”

The captions for Tables 4 and 5 should be modified to describe the numbers provided in parentheses and what they represent.

Reviewers' comments:

Reviewer's Responses to Questions

**Comments to the Author**

1. If the authors have adequately addressed your comments raised in a previous round of review and you feel that this manuscript is now acceptable for publication, you may indicate that here to bypass the “Comments to the Author” section, enter your conflict of interest statement in the “Confidential to Editor” section, and submit your "Accept" recommendation.

Reviewer #1: All comments have been addressed

Reviewer #2: (No Response)

2. Is the manuscript technically sound, and do the data support the conclusions?

Reviewer #1: Yes

Reviewer #2: Yes

3. Has the statistical analysis been performed appropriately and rigorously? 

Reviewer #1: N/A

Reviewer #2: Yes

4. Have the authors made all data underlying the findings in their manuscript fully available?

Reviewer #1: Yes

Reviewer #2: Yes

5. Is the manuscript presented in an intelligible fashion and written in standard English?

Reviewer #1: Yes

Reviewer #2: Yes

6. Review Comments to the Author

Reviewer #1: The authors addressed the comments that were raised and the manuscript has been improved. Please specify in the abstract the reason for massive exclusion of papers and clarify the use in the McMaster Critical Appraisal Tool.

Reviewer #2: The manuscript is vastly improved and the authors addressed most of the critique provided by the reviewers. This includes providing more information on the methodology used.

However, the classifications in table 5 and interpretations, as raised by reviewer 2, remain unclear. In the methods section, lines 162-167 the authors define good and poor as follows:

Parents’ level of knowledge and practice were categorized as either ‘good’ or ‘poor’ based on Bloom’s cut-off point in line with several KAP studies [26,27]. Parents with a knowledge score of more than 60% were classified as possessing good knowledge, while those parents who had a score of less than 60% were classified as having poor knowledge. As for practice, parents scoring 80% and above were categorized as demonstrating good practice,

while those parents who had a score less than 80% were classified as displaying poor practice.

However, in table 5, this is not clear – for example – study 1 it appears that 78.2% had poor knowledge, 18.2 had good knowledge and 3.6 excellent knowledge. Meaning, the authors provided the distribution and not a category based on above or below 60%. In study 2, “parents with blind…. Have knowledge about eye problems 24.7%” meaning poor? Again in study 3, “Parents know about their child’s eeye problems 89.5% and 10% do not know.” This is a distribution. The next line is “parents know where to go for eye examinations 98.2%.” Does this mean they have a knowledge score above 60% or 98.2% know where to go?

Study 4 does not have percentages at all.

Study 9, knowledge – “parents believe that routine eye exam is not necessary – this has no score.”

This confusion between the methodology and results presented in table 5 make the results section from lines 218 and on very hard to understand.

7. PLOS authors have the option to publish the peer review history of their article (what does this mean?). If published, this will include your full peer review and any attached files.

Reviewer #1: **Yes: **Hadas Ben-Eli

Reviewer #2: No

---

## [Author Response · Author response to Decision Letter 1]

26 Aug 2024

Response to Reviewers

1 Some minor additional comments that will improve the manuscript:

Abstract: parents have a “huge” responsibility – perhaps “paramount” is a better term? 

Thank you for the comment. The term “huge” has been changed to “paramount” to better reflect the critical importance of the responsibility parents have in managing their children’s eye health. 

2 Line 57: Not sure that the word “Shockingly” is required and appropriate for a scientific report. Perhaps better rephrased. Amongst the 36 million individuals worldwide affected by vision impairment as reported by the World Health Organization (WHO), 29% are children, and this figure will rise [2]. 

Thank you for the comment. The word “shockingly” has been removed, and this statement has been amended as suggested.

3 Line 230: that parents are knowledgeable 

Thank you for the comment. This statement has been amended.

4 Study 4 does not have percentages at all.

Thank you for the comment. Study 4 is a qualitative study that uses semi-structured interviews, therefore, percentages were not provided by authors. This has been stated in table 5.

5 Study 9, knowledge – “parents believe that routine eye exam is not necessary – this has no score.” 

Thank you for the comment. In Table 5, it has been specified that more than 57% of parents believe routine eye exams are not necessary. 

6 The captions for Tables 4 and 5 should be modified to describe the numbers provided in parentheses and what they represent. 

Thank you for the comment. The captions for Table 4 and 5 have been updated to clearly describe the number provided in parentheses and what they represent.

Reviewer 2

1 The manuscript is vastly improved and the authors addressed most of the critique provided by the reviewers. This includes providing more information on the methodology used.

However, the classifications in table 5 and interpretations, as raised by reviewer 2, remain unclear. In the methods section, lines 162-167 the authors define good and poor as follows:

Parents’ level of knowledge and practice were categorized as either ‘good’ or ‘poor’ based on Bloom’s cut-off point in line with several KAP studies [26,27]. Parents with a knowledge score of more than 60% were classified as possessing good knowledge, while those parents who had a score of less than 60% were classified as having poor knowledge. As for practice, parents scoring 80% and above were categorized as demonstrating good practice, while those parents who had a score less than 80% were classified as displaying poor practice. However, in table 5, this is not clear – for example – study 1 it appears that 78.2% had poor knowledge, 18.2 had good knowledge and 3.6 excellent knowledge. Meaning, the authors provided the distribution and not a category based on above or below 60%. In study 2, “parents with blind…. Have knowledge about eye problems 24.7%” meaning poor? Again in study 3, “Parents know about their child’s eeye problems 89.5% and 10% do not know.” This is a distribution. The next line is “parents know where to go for eye examinations 98.2%.” Does this mean they have a knowledge score above 60% or 98.2% know where to go? 

Thank you for your details feedback. In the revised manuscript the categorization of knowledge and practices based on Bloom’s cut-off points had been deleted because it has caused confusion. The percentages reported in Table 5 represent parents and not the score of knowledge and practices. Thus, we do not used Bloom’s cut-off point in Table 5. Table 5 in this revised manuscript reported the actual findings of percentages of parents from the selected studies.

2 Study 4 does not have percentages at all.

Thank you for the comment. Study 4 is a qualitative study that uses semi-structured interviews, therefore, percentages were not provided by authors. This has been stated in table 5.

3 Study 9, knowledge – “parents believe that routine eye exam is not necessary – this has no score.” 

Thank you for the comment. In Table 5, it has been specified that more than 57% of parents believe routine eye exams are not necessary.

---

## [Decision Letter · Decision Letter 2]

10 Sep 2024

PONE-D-23-40902R2Parents’ knowledge and practices of child eye health care: A scoping reviewPLOS ONE

Dear Dr. Mohamad Fadzil,

Thank you for submitting your manuscript to PLOS ONE. After careful consideration, we feel that it has merit but does not fully meet PLOS ONE’s publication criteria as it currently stands. Therefore, we invite you to submit a revised version of the manuscript that addresses the points raised during the review process. Please submit your revised manuscript by Oct 24 2024 11:59PM. If you will need more time than this to complete your revisions, please reply to this message or contact the journal office at plosone@plos.org. Please include the following items when submitting your revised manuscript:A rebuttal letter that responds to each point raised by the academic editor and reviewer(s). You should upload this letter as a separate file labeled 'Response to Reviewers'.A marked-up copy of your manuscript that highlights changes made to the original version. You should upload this as a separate file labeled 'Revised Manuscript with Track Changes'.An unmarked version of your revised paper without tracked changes. You should upload this as a separate file labeled 'Manuscript'.If applicable, we recommend that you deposit your laboratory protocols in protocols.io to enhance the reproducibility of your results. Protocols.io assigns your protocol its own identifier (DOI) so that it can be cited independently in the future. For instructions see: https://journals.plos.org/plosone/s/submission-guidelines#loc-laboratory-protocols. Additionally, PLOS ONE offers an option for publishing peer-reviewed Lab Protocol articles, which describe protocols hosted on protocols.io. Read more information on sharing protocols at https://plos.org/protocols?utm_medium=editorial-email&utm_source=authorletters&utm_campaign=protocols.

We look forward to receiving your revised manuscript.

Kind regards,

Liat Gantz, PhD

Academic Editor

PLOS ONE

Journal Requirements:

Additional Editor Comments:

The authors have sufficiently addressed the reviewers' past comments. However, there are still some minor comments to address before the paper can be accepted for publication as listed below:

Line 106 of this scoping review is to identify

WAS to identify

Table 1 Treatment capitol T

Following the PRISMA ScR 178 guideline, only studies relevant to the research question were included while screening titles and abstracts.

This sentence should be moved so it precedes the following:

Upon examining titles and abstracts, the majority (204) of the studies were excluded, leaving 15 studies for thorough evaluation through full-text review.

And this sentence is superfluous and can be deleted:

A significant number of studies were excluded from this review due to their lack of relevance in the title and abstract.

Line 247 were undertaken in developed countries (England and XXX) [17,20].

Replace done with undertaken and add the second country aside from England

Table 5 asterisk: *This was a qualitative study, percentages are not available.

Asterisk: developed country

Line 273: practice instead of practices

Line 279: Overall, this review

Line 299: half of the parent

Line 301: these matters instead of such matters

Line 303: suggest

Line 304: comma after income

Line 312: motivates instead of motivated

Line 314: Based on instead of according to

Also- some of the studies instead of the selected studies

Lines 324-325: necessity of seeking proper treatment options and weighing potential risks and benefits for their children’s eye health care.

Line 332” In this review, more than half of the selected studies

Line 334: period instead of comma after the references [18.20]

Line 334: Five of the selected studies reported

Line 369: are unnecessary

Line 393: Limitations (plural)

Some studies found that parents are knowledgeable and some found stigmas and lack of knowledge. Can the authors please address if there is a relationship between the knowledge vs. stigma and lack of knowledge and the area in the world (developed vs. undeveloped)?

Reviewers' comments:

Reviewer's Responses to Questions

**Comments to the Author**

1. If the authors have adequately addressed your comments raised in a previous round of review and you feel that this manuscript is now acceptable for publication, you may indicate that here to bypass the “Comments to the Author” section, enter your conflict of interest statement in the “Confidential to Editor” section, and submit your "Accept" recommendation.

Reviewer #1: All comments have been addressed

2. Is the manuscript technically sound, and do the data support the conclusions?

Reviewer #1: Yes

3. Has the statistical analysis been performed appropriately and rigorously? 

Reviewer #1: N/A

4. Have the authors made all data underlying the findings in their manuscript fully available?

Reviewer #1: Yes

5. Is the manuscript presented in an intelligible fashion and written in standard English?

Reviewer #1: Yes

6. Review Comments to the Author

Reviewer #1: The authors addressed all the comments that I have referred to them in this paper. I recommend to accept this paper for publication in PLosOne journal.

7. PLOS authors have the option to publish the peer review history of their article (what does this mean?). If published, this will include your full peer review and any attached files.

Reviewer #1: No

---

## [Author Response · Author response to Decision Letter 2]

13 Sep 2024

1 The authors have sufficiently addressed the reviewers' past comments. However, there are still some minor comments to address before the paper can be accepted for publication as listed below:

Line 106 of this scoping review is to identify

WAS to identify 

Thank you for the comment. This statement has been amended.

2 Table 1 Treatment capitol T 

Thank you for the comment. This has been amended.

3 Following the PRISMA ScR 178 guideline, only studies relevant to the research question were included while screening titles and abstracts.

This sentence should be moved so it precedes the following:

Upon examining titles and abstracts, the majority (204) of the studies were excluded, leaving 15 studies for thorough evaluation through full-text review.

And this sentence is superfluous and can be deleted:

A significant number of studies were excluded from this review due to their lack of relevance in the title and abstract. 

Thank you for your comment. The sentences have been corrected as suggested.

4 Line 247 were undertaken in developed countries (England and XXX) [17,20]. 

Thank you for the comment. This statement has been amended. The two studies were undertaken in England only. So, now the sentence reads as below:

The majority of the studies were conducted in developing countries – India [13,16], Nigeria [7,9,21], Saudi Arabia [11,15], Sudan [18], Swaziland [19] , and only two (n=2) studies were undertaken in a developed country (England) [17,20].

5 Table 5 asterisk: *This was a qualitative study, percentages are not available.

Asterisk: developed country 

Thank you for the comment. In Table 5, it has been amended; asterisk is to represent developed country. A bracket has been added to the sentence “This was a qualitative study, percentages are not available” 

6 Line 273: practice instead of practices 

Thank you for your comment. The word has been corrected as suggested.

7 Line 279: Overall, this review 

Thank you for the comment. This statement has been amended.

8 Line 299: half of the parent 

Thank you for the comment. This statement has been amended.

9 Line 301: these matters instead of such matters 

Thank you for the comment. This statement has been amended.

10 Line 303: suggest 

Thank you for your comment. The word has been corrected as suggested.

11 Line 304: comma after income 

Thank you for the comment. The comma has been added.

12 Line 312: motivates instead of motivated 

Thank you for your comment. The word has been corrected as suggested.

13 Line 314: Based on instead of according to 

Also- some of the studies instead of the selected studies 

Thank you for the comment. This statement has been amended as suggested.

14 Lines 324-325: necessity of seeking proper treatment options and weighing potential risks and benefits for their children’s eye health care. 

Thank you for the comment. This statement has been amended as suggested.

15 Line 332” In this review, more than half of the selected studies 

Thank you for the comment. This statement has been amended as suggested.

16 Line 334: period instead of comma after the references [18.20] 

Thank you for your comment. This has been amended.

17 Line 334: Five of the selected studies reported 

Thank you for the comment. This statement has been amended as suggested.

18 Line 369: are unnecessary 

Thank you for your comment. This has been amended.

19 Line 393: Limitations (plural)

 Thank you for your comment. This has been amended.

20 Some studies found that parents are knowledgeable and some found stigmas and lack of knowledge. Can the authors please address if there is a relationship between the knowledge vs. stigma and lack of knowledge and the area in the world (developed vs. undeveloped)? 

Thank you for your comment. Explanation regarding the relationship between knowledge and stigma and the are in the world have been added in discussion (lines 366-375).

---

## [Editor Report · Decision Letter 3]

22 Oct 2024

Parents’ knowledge and practices of child eye health care: A scoping review

PONE-D-23-40902R3

Dear Dr. Fadzil,

We’re pleased to inform you that your manuscript has been judged scientifically suitable for publication and will be formally accepted for publication once it meets all outstanding technical requirements.

Kind regards,

Liat Gantz, PhD

Academic Editor

PLOS ONE

Additional Editor Comments (optional):

The reviewer's comments have been adequately addressed.
---

## [Editor Report · Acceptance letter]

5 Nov 2024

PONE-D-23-40902R3 

PLOS ONE

Dear Dr. Mohamad Fadzil, 

I'm pleased to inform you that your manuscript has been deemed suitable for publication in PLOS ONE. Congratulations! Your manuscript is now being handed over to our production team.

Kind regards, 

on behalf of

Dr. Liat Gantz 

Academic Editor

PLOS ONE